

# Age of onset of cannabis use and decision making under uncertainty

Jose Ramón Alameda-Bailén[1], Pilar Salguero-Alcañiz[1],*,
Ana Merchán-Clavellino[2],* and Susana Paíno-Quesada[3],*

[1] Basic Psychology, University of Huelva, Huelva, Spain
[2] Basic Psychology, University of Cádiz, Puerto Real, Cádiz, Spain
[3] Personality, Evaluation and Psychological Treatments, University of Huelva, Huelva, Spain
* These authors contributed equally to this work.

Corresponding author
Jose Ramón Alameda-Bailén,
alameda@uhu.es

## ABSTRACT

**Objective:** Cannabis, like other substances, negatively affects health, inducing respiratory problems and mental and cognitive alterations. Memory and learning disorders, as well as executive dysfunctions, are also neuropsychological disorders associated to cannabis use. Recent evidence reveals that cannabis use during adolescence may disrupt the normal development of the brain. This study is aimed to analyze possible differences between early-onset and late-onset cannabis consumers.

**Method:** We used a task based on a card game with four decks and different programs of gains/losses. A total of 72 subjects (19 women; 53 men) participated in the study; they were selected through a purposive sampling and divided into three groups: early-onset consumers, late-onset consumers, and control (non-consumers). The task used was the "Cartas" program (computerized version based on the Iowa Gambling Task (IGT)), with two versions: direct and inverse. The computational model "Prospect Valence Learning" (PVL) was applied in order to describe the decision according to four characteristics: utility, loss aversion, recency, and consistency.

**Results:** The results evidence worst performance in the IGT in the early-onset consumers as compared to late-onset consumers and control. Differences between groups were also found in the PVL computational model parameters, since the process of decision making of the early-onset consumers was more influenced by the magnitude of the gains-losses, and more determined by short-term results without loss aversion.

**Conclusions:** Early onset cannabis use may involve decision-making problems, and therefore intervention programs are necessary in order to reduce the prevalence and delay the onset of cannabis use among teenagers.

# INTRODUCTION

According to the Spanish Observatory on Drugs and Drug Addiction (2015), cannabis is the most frequently consumed illegal substance in Spain, and the age of onset is also the earliest (18.6 years). The use of cannabis is considerably more extended among youth,

and the highest incidence rates occur between 15 and 17 years, reaching 47.0% (*Observatorio Español de la Droga y las Toxicomanías, 2015*). As the age of the population increases, the use cannabis decreases.

Cannabis, like other substances, negatively affects health, inducing respiratory problems and mental and cognitive alterations (psychosis, anxiety, dependence, etc.). Memory and learning disorders (*Crane et al., 2013*; *Jager et al., 2006*), and executive dysfunctions (*Alameda-Bailén, Paíno-Quesada & Mogedas-Valladares, 2012*; *Alameda-Bailén et al., 2014*; *Crean, Crane & Mason, 2011*; *Fontes et al., 2011*), are also neuropsychological disorders related to cannabis use. There is some consensus about the determinants of the short and long-term consequences of cannabis use: age of onset of consumption, consumption frequency, consumption duration, the amount of tetrahydrocannabinol in the cannabis, etc. (*Volkow et al., 2016*). Among these factors, the age of onset of consumption is currently the most relevant, because the most recent empirical evidence reveals that cannabis use during adolescence may disrupt the normal development of the brain. Specifically, cannabis consumption during this period seems to affect specific brain maturation processes, such as synaptic pruning and the development of white matter, and therefore the alterations in these processes could lay the foundation for cognitive and mental health impairments in adulthood (*Lubman, Cheetham & Yücel, 2015*).

These results are quite similar to those of other works that highlight that the age of cannabis use onset and the resulting consumption patterns contribute to the risk of suffering from mental illnesses, for example psychosis. In this respect, *Di Forti et al. (2014)* observed that there are substantial differences in the age of psychosis onset among consumers who began to use cannabis before age 15 and after that age.

There are different studies that show how the age of onset of consumption affects cognitive processes (*Gruber et al., 2012*; *Pope et al., 2003*; *Wilson et al., 2000*). *Fontes et al. (2011)* highlighted that the early-onset group had worse performance than the late-onset group in attention tasks, impulse control, and executive function, and thus it can be concluded that regular consumption before age 15 can have adverse effects on neurocognitive functioning.

On another hand, *Meier et al. (2012)* showed in a longitudinal study that the greatest neuropsychological decline occurred in the adolescent-onset cannabis users, whose cognitive functions were not restored when they ceased consumption. The results of *Gruber et al. (2012)* are congruent with these findings, as a worse performance in the executive functions (WAIS-R, Stroop, Wisconsin, etc.) was observed in cannabis consumers with onset before the age of 16.

To sum up, evidence seems to support the idea that adolescence is a period of special risk, as cannabis use at this stage of development causes adverse reactions that are more severe and persistent than those related to consumption during adulthood. The neurotoxic damage of cannabis in the teenager brain could produce in these early-onset consumers irreversible alterations to the mental health and the cognitive functions.

Among the cognitive functions impaired by cannabis use, the decision-making processes are the most studied (*Alameda-Bailén, Paíno-Quesada & Mogedas-Valladares, 2012*; *Alameda-Bailén et al., 2014*; *Mogedas-Valladares & Alameda-Bailén, 2011*).

Such processes highlights the somatic marker hypothesis (*Damasio, 1994*), a research paradigm in which all the variations of the decision-making processes of the different communities are evaluated through the Iowa Gambling Task (IGT; *Bechara et al., 1994*). IGT simulates a card game that allows studying decision-making processes in uncertain situations of everyday life in the laboratory. Healthy individuals make more advantageous choices, however, orbitofrontal damaged patients and people with addictions, among others, tend to choose cards from the most disadvantageous decks and cannot anticipate the consequences of their choices because of their difficulties to develop emotional signals related to the affective value of the different choices (*Alameda-Bailén, Paíno-Quesada & Mogedas-Valladares, 2012*; *Bechara, 2003*; *Bechara & Damasio, 2002*; *Bechara et al., 1997*; *Bechara, Tranel & Damasio, 2000*; *Mogedas-Valladares & Alameda-Bailén, 2011*). In people with addictions, these difficulties to anticipate the consequences of their choices can be a consequence of drugs use, but also the cause of initiating consumption. According to *Bechara, Tranel & Damasio (2000)* and *Gordillo et al. (2010)*, the results of the IGT may be due to three factors:

Hypersensitivity to reward: the perspective of obtaining a large immediate reward is higher than any chance of loss in the future, and preference for the advantageous decks is related to high response times.

Insensitivity to punishment: the perspective of a large loss does not invalidate any chance of obtaining some reward, and preference for the advantageous decks is related to low response times.

Insensitivity to future consequences: which is a guideline based on immediate perspectives, and preference for the disadvantageous decks is related to normal response times.

To determine which of these options is more likely, *Bechara, Tranel & Damasio (2000)* proposed a version of IGT in which losses now turn into gains, and vice-versa; that is, the gains/losses structure is reversed. Preference for advantageous decks in the reverse task is coherent with sensitivity to punishment and hypersensitivity to reward, depending on the levels of activation associated with losses and gains, respectively. However, the preference for the disadvantageous decks is consistent with insensitivity to consequences, as defended by *Bechara, Tranel & Damasio (2000)*, and higher response times, which is related to high levels of emotional activation. Although there are models (not based in IGT) that emphasize the importance of reward hypersensitivity, they reflect a pre-existing vulnerability for addictive behaviors (*Alloy et al., 2009*; *Nusslock & Alloy, 2017*), and therefore reward hypersensitivity should lead to greater substance use and prospectively put an individual at risk for addiction.

Performance on the IGT can be analyzed by computational cognitive models, such as the Prospect Valence–Learnig, (PVL) (*Ahn et al., 2008*, *2011*, *2014*), complementarily to the Gambling Index, which enable defining the cognitive mechanisms involved in the decision-making by four parameters (utility rule, loss aversion, recency, and consistency). The PVL model, based on Bayesian logic, has three general assumptions (*Ahn et al., 2008*):

The evaluation of the positive/negative results can be represented by a one-dimensional utility function.

Expectancies about each deck are learned by what is experienced in each trial. These expectancies determine the choice probabilities of each deck on each trial.

The four PVL parameters and the equations to calculate them, are:
To rate a card:

$$u(t) = \begin{cases} x(t)x^{\alpha} \rightarrow \text{if } x(t) \geq 0 \\ -\lambda|x(t)|^{\alpha} \rightarrow \text{if } x(t) < 0 \end{cases}$$

where:

–*Utility Rule* or **Reward sensitivity (α)**. This value regulates the shape of the utility function. High values of α indicate that the person is more sensitive to feedback outcomes, whereas low values of α indicate low sensitivity to feedback outcomes.

–**Loss aversion (λ)**. This value determines sensitivity to losses compared to gains. A $\lambda$ value less than 1 indicates more sensitivity to gains than to losses, whereas a $\lambda$ value greater than 1 indicates more sensitivity to losses than to gains.

To create deck expectancy ($E$), for deck $j$ on trial $t$, the equation is:

$$E_j(t) = A \cdot E_j(t-1) + \delta_j(t) \cdot u(t)$$

Where $j$ refers to deck A, B, C, or D. $\delta_j(t)$ is a dummy variable equal to 1 if deck $j$ was chosen on trial $t$, and otherwise is 0. $A$ is the recency or learning rate parameter.

–**Recency parameter ($A$)**. Determines how much the past expectancy is discounted. When $A$ is close to 0, more weight is granted to past outcomes, and when $A$ is close to 1, more weight is granted to recent outcomes.

The equation to calculate the probability of choosing Deck $j$ is:

$$Pr[D(t+1) = j] = \frac{e^{\theta(t) \cdot e_j(t)}}{\sum\limits_{K=1}^{4} e^{\theta(t) \cdot E_k(t)}}$$

To calculate the consistency between choices and expectancies, the equation is:

$$\theta(t) = 3^c - 1$$

–**Consistency or Response Sensitivity ($c$)** is a consistency parameter (choice sensitivity), where low values denote a random sensitivity choice, whereas high values show a deterministic sensitivity choice.

The PVL model has been applied to clinical samples and has allowed the identification of decision-making patterns in the IGT (*Ahn et al., 2008*, *2011*, *2014*; *Alameda-Bailén, Paíno-Quesada & Mogedas-Valladares, 2012*; *Alameda-Bailén et al., 2014*, *2017*).

Based on this background, this study aimed to analyze possible differences between early-onset and late-onset cannabis consumers and compare performance of each group in the PLV computational model of the IGT. As discussed earlier, different studies show how the age of onset of consumption affects cognitive processes, especially with early-onset, which shows a worse task performance, impulse control or executive function. Therefore, considering that decision making is part of the executive functions, we expect to find a lower number of advantageous choices in early-onset consumers as compared to late-onset consumers and controls. Early-onset consumers could take more risky decisions.

## METHOD

### Participants

All the participants were degree, Master's or Doctoral students, as well as University staff. We collected data on the consumption of cannabis and other drugs in a group of 75 volunteers, to differentiate between the early and late onset cannabis use. A total of 18 participants reported an age of onset of 16 years or less, and from this age we established the size of the late onset and control group. The groups were matched by sex and education level, with no significant differences in these variables. A total of 19 women and 53 men, aged between 18 and 36 years, participated in this study. They were divided into three groups ($n = 18$, early-onset cannabis users; $n = 18$, late-onset cannabis users; $n = 36$, non-consumers as control group). Participants had not history of gambling problems, consumption of alcohol or other significant drugs, and they were not under psychopharmacological treatment. All participants provided informed consent to participate in the study. The procedure was carried out in accordance with the recommendations of ethics guidelines of the University of Huelva and according to the Declaration of Helsinki.

Since the groups were classified by age, we included the condition that the years of consumption and the daily consume of cannabis cigarettes were not significantly different, otherwise the early-onset consumers would always show more years of consumption than the late-onset consumers.

The characteristics of the groups were:

*Early-onset cannabis consumers*: 18 participants (five women; 13 men), aged between 18 and 24 years ($M = 21$, SD = 2.01). Four participants had elementary studies, five had secondary education and nine were university students. The mean duration of consumption was 5.66 years (SD = 2.40), and consumption was 4.72 cannabis cigarettes per day (SD = 2.89). We selected participants with an age of onset of cannabis use between 14 and 16 years ($M = 15.3$, SD = 0.88).

*Late-onset cannabis consumers*: 18 participants (four women; 14 men), aged between 21 and 36 years ($M = 27.55$, SD = 3.80). Five participants had elementary studies, five had secondary education and eight were university students. The mean duration of consumption was 4.16 years (SD = 2.70), and consumption was 3.83 cannabis cigarettes per day (SD = 2.75). There were no significant differences either in the duration of consumption or in the amount consumed between the consumer groups ($p > 0.05$).

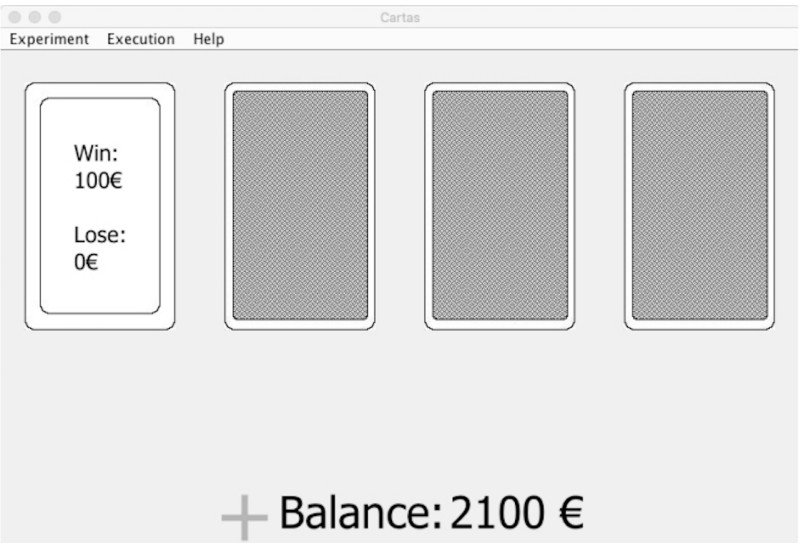

**Figure 1  Screenshot of the Cards program.**     

We selected participants with an age of onset of cannabis use between 20 and 28 years ($M$ = 23.3, SD = 3.12).

*Control group*: 36 participants (10 women; 26 men), aged between 19 and 36 years ($M$ = 24.69, SD = 4.21). Nine participants had elementary studies, eleven had finished secondary education, and sixteen were university graduates. As the mean age of the consumption groups (early/late-onset) was different, the control group had twice the number of participants to match the ages of each group. In addition, we found that there were no significant differences in any of the study variables, based on the age of the control group.

## Task

The software "Cartas" (*Palacios, Paíno & Alameda, 2010*), a computerized version of the Iowa Gambling Task (*Bechara et al., 1994*), was used to assess the decision-making process. The "Cartas" task simulates a card game in which the individual must choose from four decks across different trials (Fig. 1; Table 1).

After exploring decks, normally we continue choosing the decks that can provide large gains, although there may be larger losses (approximately, in 30 plays out of 100), until they switch the strategy and start to select decks that are more beneficial at the long-term, that is, those with less gain but also with less loss (*Martínez-Selva et al., 2006*). The task starts with an imaginary 2,000€, and the participants must try to either increase this amount or retain it, across 100 plays. A reverse task (EFGH version), in which A-B decks become advantageous and C-D decks become disadvantageous, was also used in the present study. In the reverse version, the game is based on frequent losses and occasional gains.

## Procedure

The participants were informed of the objectives of the study and they voluntarily participated in a single individual 15–20-min session. Before starting the task, data about
**Table 1 Gains-loss program and the probability of loss ($p^*$) of each deck in the IGT in a cycle of 10 plays.**

| Deck | Iowa Gambling Task (IGT) | | |
|---|---|---|---|
| | Gain | Loss | $p^*$ |
| A (disadvantageous) | 100 | 150 | 0.5 |
| | | 300 | |
| | | 200 | |
| | | 250 | |
| | | 350 | |
| B (disadvantageous) | 100 | 1,250 | 0.1 |
| C (advantageous) | 50 | 25 | 0.5 |
| | | 25 | |
| | | 50 | |
| | | 75 | |
| | | 75 | |
| D (advantageous) | 50 | 250 | 0.1 |

Note:
Decks A and B provide the highest short-term gains and the highest long-term losses (higher risk). Decks C and D provide little money at the short term but higher.

age, sex, educational level, and drug use (onset age and the daily amount consumed) were collected. Participants performed both the classic and the reverse version of the task. This procedure was conducted in a room provided with all the necessary tools to the study. After completion the study, an explanation of the obtained results was provided to all participants under request. Participants were informed that the task objective was to increase start-up money, but they would not be rewarded with real money, as data show that there is no difference in IGT performance when using real and fictitious money (*Bowman & Turnbull, 2004*), or when participants can keep all or part of the gains (*Bickel et al., 2009*; *Carter & Pascualini, 2004*; *Fernie & Tunney, 2006*; *Schmitt, Brinkley & Newman, 1999*; *Suzuki et al., 2003*).

## Data analysis

SPSS software was used to analyze data of the PVL model of IGT.

The dependent variables used were:

Percentage of advantageous/disadvantageous choices: Choices made in 100 trials, and the partial percentages in five blocks of 20 trials (blocks B1: trial 1–20, B2: 21–40, B3: 41–60, B4: 61–80, & B5: 81–100), both in the classic and the reverse task.

The number of cards chosen in each deck (A, B, C, D) in both versions of the task.

The time spent in the selection of the different groups for the advantageous and disadvantageous choices in the reverse version options.

The PVL parameters: α (*utility rule*), λ (*loss aversion*), A (*recency*), c (*consistency*) were calculated through a R script, using the maximum likelihood method, MLE (*Ahn et al., 2008*), and the decay-rule (*Erev & Roth, 1998*), as it consistently shows better

models of post-hoc fit than the delta-rule (*Rescorla & Wagner, 1972*) with respect to the IGT (*Yechiam et al., 2005*).

The independent variable used was the age at onset of cannabis use. The influence of variables such as sex, age, educational level, daily consumed cannabis cigarettes, and years of consumption was also analyzed. To analyze the influence of consumption on task performance and the parameters of the PVL model, Student's *t*-tests for independent samples were calculated, comparing the results according to cannabis use. To compare the evolution across the different blocks in each group, repeated measures analyses of variance were used.

# RESULTS

## Normal procedure

The control group made advantageous choices on 60.94% (SD = 5.65, 95% CI [57.8–64.1]) in the total IGT (IG = 21.34), and the late-onset group made 49.78% (SD = 13.07, 95% CI [45.3–54.2] IG = −0.44) but there were no significant differences ($t$ (1, 17) = 0.072; $p$ = 0.943, 95% CI [12.5–13.4]). The early-onset group made advantageous choices on 32.78% (SD = 11.22, 95% CI [28.3–37.2] IG = −34.44), the differences between this group and the control group ($t$ (1, 35) = 11.619; $p$ = 0.000, 95% CI [18.1–25.7]) and the late-onset group ($t$ (1, 17) = 6.508; $p$ = 0.000, 95% CI [23.2–45.6]) were significant. Figure 2 shows the advantageous/disadvantageous choices of the different groups.

The between-group differences observed in the percentages of elections were significant ($F$ (2, 71) = 53.48, $p$ = 0.000, $\eta_p^2$ = 0.608). Bonferroni-corrected paired comparisons also showed significant differences between the control and early-onset groups ($p$ = .000, 95% CI [21.5–34.9]), the control and late-onset groups ($p$ = 0.000, 95% CI [45.0–17.9]), and the late-onset and early-onset groups ($p$ = 0.000, 95% CI [9.3–24.7]).

In the block analysis, the advantageous choices of the early-onset group were lower than the disadvantageous choices throughout the five blocks (Table 2; Fig. 3), whereas the advantageous choices were detected in the third block in the late-onset group, with an increasing trend to such choices. The control group showed an increasing trend to advantageous choices in the second block.

The analyses of the advantageous choices in the different groups throughout the blocks (B1–B5) (Table 2; Fig. 3) showed significant differences between the early-onset consumers and the other two groups in the B1. In the B2 there were significant differences between the control group and the early-onset consumers. In the B3 there were significant differences between the early-onset consumers and the control group, and between the early-onset and late-onset consumers. In the B4 there were significant differences between both groups of consumers, and between the control and consumer groups (early and late onset). In the B5, significant differences were observed between consumers and the control group.

Repeated measures ANOVA revealed differences in the scores of the different blocks ($F$ (4, 68) = 13,943, $p$ = 0.000, $\eta_p^2$ = 0.451). Analyzing each group independently, the early-onset group showed significant differences between blocks ($F$ (4, 14) = 4.546;

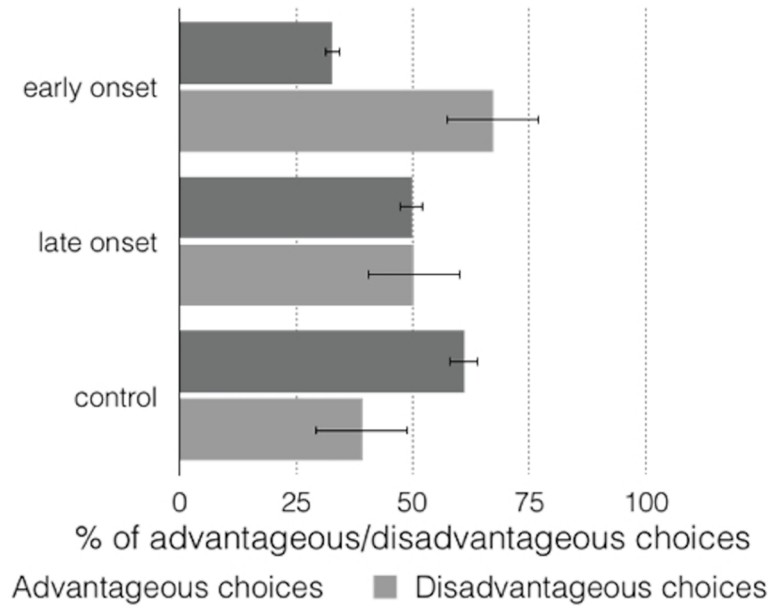

early onset

late onset

control

0    25    50    75    100

% of advantageous/disadvantageous choices

■ Advantageous choices        ■ Disadvantageous choices

**Figure 2  Graph of advantageous/disadvantageous choices in total task.**

**Table 2  Descriptive and statistical analysis of the percentage of the number of selections from the advantageous decks in different blocks.**

| | Early onset (EO) | | | Late onset (LO) | | | Control (C) | | | EO-LO | | EO-C | | LO-C | |
|---|---|---|---|---|---|---|---|---|---|---|---|---|---|---|---|
| | | | | | | | | | | $df$ (1, 34) | | $df$ (1, 52) | | $df$ (1, 52) | |
| | $M$ | SD | Range | $M$ | SD | Range | $M$ | SD | Range | $t$ | $p^*$ | $t$ | $p^*$ | $t$ | $p^*$ |
| B1 | 26.39 | 15.32 | 10–55 | 40.83 | 18.01 | 10–70 | 43.61 | 16.15 | 10–80 | 2.592 | 0.031 | 3.756 | 0.002 | 0.573 | 1 |
| B2 | 35.28 | 17.61 | 15–80 | 46.94 | 20.73 | 05–95 | 59.44 | 19.45 | 00–100 | 1.820 | 0.224 | 4.437 | 0.000 | 2.178 | 0.085 |
| B3 | 28.61 | 15.51 | 05–65 | 51.94 | 23.77 | 05–100 | 61.53 | 21.51 | 00–100 | 3.488 | 0.004 | 5.774 | 0.000 | 1.490 | 0.346 |
| B4 | 32.50 | 15.74 | 10–60 | 55.00 | 18.79 | 35–100 | 69.44 | 19.08 | 35–100 | 3.895 | 0.001 | 7.088 | 0.000 | 2.636 | 0.023 |
| B5 | 41.11 | 17.03 | 20–70 | 55,56 | 17.17 | 30–85 | 70.69 | 15.45 | 50–100 | 2.291 | 0.057 | 6.411 | 0.000 | 3.571 | 0.002 |

Note:
* The $p$-value has been corrected by the Bonferroni adjustment for multiple comparisons.

$p = 0.015$, $\eta_p^2 = 0.565$), and Bonferroni-corrected comparisons showed significant differences between B1 and B5 ($p = 0.010$, 95% CI [3.0–29.2]). Comparisons revealed that there were no differences in the late-onset consumers ($F$ (4, 14) = 1.804; $p = 0.184$, $\eta_p^2 = 0.340$) and therefore the slight increasing trend was not confirmed in this group. Finally, in the control group, significant differences were observed across the blocks ($F$ (4, 32) = 11.623; $p = 0.000$, $\eta_p^2 = 0.592$). Bonferroni-corrected comparisons revealed significant differences between B1 and the other blocks (B1–B2, $p = 0.008$, 95% CI [3.3–32.2]); B1–B3, $p = 0.011$, 95% CI [2.9–34.6]; B1–B4, $p = 0.000$, 95% CI [13.5–40.7] and B1–B5, $p = 0.000$, 95% CI [16.2–41.8]), and between B2 and B5, ($p = 0.036$, 95% CI [0.5–22.0]).

Table 3 shows the results obtained for the parameters of the PVL model, where no significant differences were found between the groups in consistency (c: $F$ (2, 69) = 0.804, $p = 0.452$), with values in each group close to 1. These results indicate little

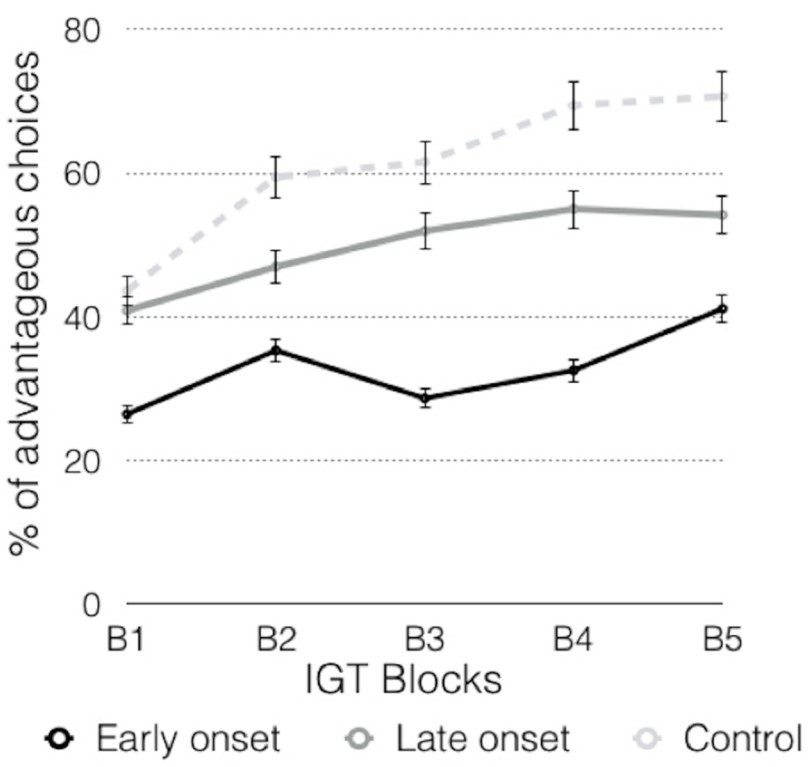

**Figure 3 Graphic representation of the advantageous choices across the blocks.**

**Table 3 Descriptive and statistical analysis of the PVL parameters.**

| | Early onset (EO) | | | Late onset (LO) | | | Control (C) | | | EO-LO | | EO-C | | LO-C | |
| | | | | | | | | | | df (1, 34) | | df (1, 52) | | df (1, 52) | |
| | M | SD | Range | M | SD | Range | M | SD | Range | t | p* | t | p* | t | p* |
| A | 0.69 | 0.38 | 0–1 | 0.38 | 0.38 | 0–1 | 0.41 | 0.39 | 0–1 | 2.420 | 0.059 | 2.452 | 0.048 | 0.288 | 1 |
| α | 0.86 | 0.24 | 0.3–1 | 0.58 | 0.43 | 0.1–1 | 0.36 | 0.35 | 0.1–1 | 2.375 | 0.064 | 5.374 | 0.000 | 2.027 | 0.094 |
| c | 1 | 1.03 | 0.1–5 | 1.57 | 1.39 | 0.1–5 | 1.45 | 1.64 | 0.1–5 | 1.399 | 0.727 | 1.047 | 0.887 | 0.279 | 1 |
| λ | 0.03 | 0.05 | 0.1–0.22 | 1.33 | 2.04 | 0.1–5 | 2.72 | 1.94 | 0.1–5 | 2.704 | 0.077 | 5.866 | 0.000 | 2.448 | 0.019 |

**Note:**
* The p-value has been corrected by the Bonferroni adjustment for multiple comparisons.

coherence between expectations and the final choice, that is, the choice of the deck was random.

There were significant differences in the factor group in loss aversion ($\lambda$: $F_{(2, 69)}$ = 15.414, $p$ = 0.000). In early-onset consumers, $\lambda$ value was close to 0, which indicates that this group perceived losses as neutral events, that is, this group showed more sensitivity to gains than to losses, and there was no loss aversion. In late-onset consumers, $\lambda$ value was close to 1, which suggests that losses and gains were considered equally, whereas in the control group, $\lambda$ value was greater than 1, indicating more sensitivity to losses than to gains, and loss aversion (Fig. 4).

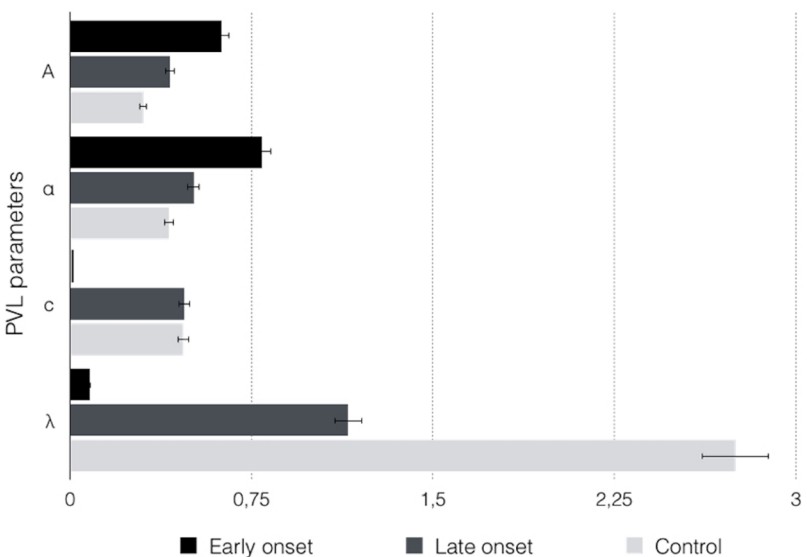

**Figure 4 Graphic representation and mean differences of PVL model parameters in the three groups of participants.**

Regarding the shape of the utility function ($\alpha$: $F(2, 69) = 12.258$, $p = 0.000$), the early-onset and late-onset consumers showed an $\alpha$ value close to 1, suggesting that their choices were more sensitive to feedback outcomes, whereas in the control group, the $\alpha$ value indicated a low sensitivity to feedback outcomes.

The early-onset consumers showed higher values in recency ($A$: $F(2, 69) = 3.743$, $p = 0.029$), close to 1, which suggests that the value of the last card had a large influence on the expectation created by that deck, and consequently all the previous choices were forgotten, and more weight was granted to recent outcomes. However, in the control group and the late-onset consumers, with $A$ scores close to 0, the value of the previous performance had little influence on the expectation created by the decks, with previous experiences prevailing, so forgetting was a slower process, and more weight was granted to past outcomes.

## Inverse procedure

The results observed in the classic version (ABCD) of the IGT may be due to hypersensitivity to reward, insensitivity to punishment, or insensitivity to future consequences (*Bechara, Tranel & Damasio, 2000*). Preference in the reverse task for the disadvantageous decks is related to insensitivity to punishment when low response times are obtained and to hypersensitivity to reward with high response times (*Gordillo et al., 2010*). Preference for the unfavorable decks is consistent with insensitivity to consequences (*Gordillo et al., 2010*), which is the option argued by *Bechara, Tranel & Damasio (2000)*.

In the total IGT-inverse performance, the control group made advantageous choices on 51% (SD = 16.70, 95% CI [45.3–56.0] IG = 2), versus 54.72% (SD = 14.54, 95% CI [47.2–62.3] IG = 9.44) of the late-onset group and 42.722% (SD = 10.77, 95% CI [34.7–49.8] IG = −14.556) of the early-onset group. Comparing the results obtained

**Table 4 Descriptive and statistical analysis of the measurements recorded in each block of the reverse task.**

| | Early onset (EO) | | | Late onset (LO) | | | Control (C) | | | EO-LO | | EO-C | | LO-C | |
| | | | | | | | | | | df (1, 34) | | df (1, 52) | | df (1, 52) | |
| | M | SD | Range | M | SD | Range | M | SD | Range | t | p* | t | p* | t | p* |
| B1 | 46.39 | 17.81 | 25–100 | 58.61 | 22.02 | 30–100 | 46.53 | 22.32 | 0–100 | 1.831 | 0.265 | 0.023 | 1 | 1.884 | 0.158 |
| B2 | 44.17 | 16.11 | 0–65 | 60.56 | 26.06 | 15–100 | 50.28 | 30.26 | 0–100 | 2.270 | 0.200 | 0.799 | 1 | 1.230 | 0.544 |
| B3 | 35.83 | 17.17 | 0–60 | 57.50 | 25.74 | 20–100 | 52.08 | 29.87 | 0–100 | 2.971 | 0.047 | 2.133 | 0.106 | 0.656 | 1 |
| B4 | 41.94 | 19.19 | 10–80 | 52.22 | 18.73 | 15–85 | 46.67 | 30.71 | 0–100 | 1.627 | 0.698 | 0.595 | 1 | 0.703 | 1 |
| B5 | 42.78 | 14.58 | 0–65 | 44.72 | 16.40 | 20–70 | 51 | 28.15 | 0–100 | 0.376 | 1 | 0.941 | 0.639 | 0.656 | 1 |

Note:
* The p-value has been corrected by the Bonferroni adjustment for multiple comparisons.

in the inverse task with the normal task, we observe that late-onset consumers improve in their advantageous choices, going from 32.78% to 42.72%, resulting in this significant increase ($t (1, 17) = 3.779$, $p = 0.001$), although again this group showed the lowest scores, predominantly with disadvantageous choices. The late-onset group also increase their advantageous choices, going from 49.78% to 54.72%, but this increase is not significant ($t (1, 17) = 1.953$, $p = 0.067$), while the control group decrease your advantageous choices, going from 60.94% to 51% ($t (1, 35) = 4.666$, $p = 0.000$).

The differences observed in the percentages of advantageous choices of inverse task were not significant ($F (2, 71) = 2.903$, $p = 0.062$, $\eta_p^2 = 0.078$). The Bonferroni-corrected comparisons showed no significant differences ($p > 0.05$).

Table 4 shows the descriptive statistics for the different blocks of the reverse task. Differences between the early-onset and late-onset consumers were found only in the B3.

In the task evolution (Fig. 5), a declining trend in the late-onset consumers was found, whereas the early-onset consumers also showed a declining trend in the first block, and an increasing trend in the last block, but when the task was completed, there were fewer advantageous choices than those made at the beginning. The control group showed an increasing tendency in the three first blocks. However, these trends were not confirmed by the repeated measures ANOVA ($F (4, 67) = 0.296$, $p = 0.880$, $\eta_p^2 = 0.017$), because no significant between-group differences were found in the evolution of the task.

With respect to the parameters of the PVL model (Table 5), no significant differences in any of the parameters were found for the different groups, indicating a similar performance in each group. Very high $\lambda$ values were obtained in all the groups, which seems to confirm loss aversion. The shape of the utility showed values tending towards 1 in all the groups, suggesting that the choices were conditioned by the gains-losses magnitude. With respect to recency ($A$), the values of all the groups were close to 0, indicating that the last choice made did not affect the next one, forgetting was more gradual, and previous experiences prevailed. The consistency parameters ($c$) revealed random choices.

Figure 6 shows the response times of the consumer groups and the control group in the IGT, with higher response times for consumers in the advantageous choices.

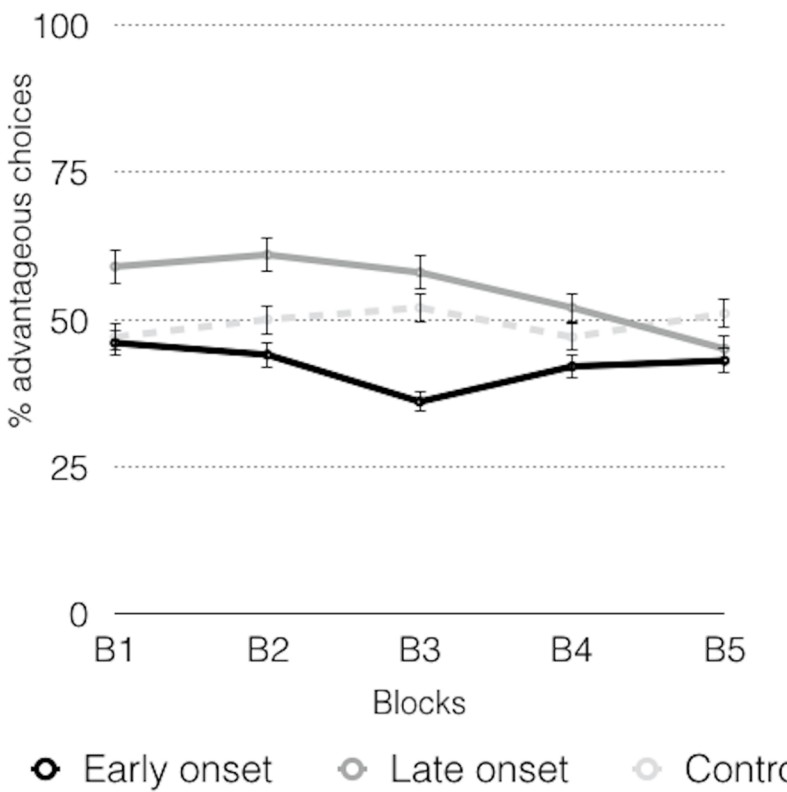

**Figure 5 Advantageous choices across the inverse task.**

**Table 5 Descriptive and statistical analysis of the PVL parameters in the reverse task.**

| | Early onset (EO) | | | Late onset (LO) | | | Control (C) | | | EO-LO | | EO-C | | LO-C | |
|---|---|---|---|---|---|---|---|---|---|---|---|---|---|---|---|
| | | | | | | | | | | df (1, 34) | | df (1, 52) | | df (1, 52) | |
| | M | SD | Range | M | SD | Range | M | SD | Range | t | p* | t | p* | t | p* |
| A | 0.26 | 0.269 | 0–1 | 0.47 | 0.35 | 0–1 | 0.35 | 0.32 | 0–1 | 2.058 | 0.133 | 1.108 | 0.878 | 1.250 | 0.589 |
| α | 0.75 | 0.364 | 0–1 | 0.63 | 0.43 | 0–1 | 0.73 | 0.37 | 0–1 | 0.913 | 1 | 0.114 | 1 | 0.973 | 0.979 |
| c | 0.68 | 1.12 | 0–5 | 0.93 | 1.13 | 0–4 | 0.98 | 0.98 | 0–5 | 0.655 | 1 | 0.994 | 1 | 0.165 | 1 |
| λ | 3.04 | 2.15 | 0–5 | 2.33 | 2.09 | 0–5 | 2.22 | 2.15 | 0–5 | 1.008 | 1 | 1.318 | 0.568 | 0.169 | 1 |

**Note:**
* The p-value has been corrected by the Bonferroni adjustment for multiple comparisons.

No significant differences were found respect to the sex, age, educational level, daily consumption and years of consumption factors, in both versions of the task.

## DISCUSSION

In the classic procedure of the task, cannabis consumers obtained worse results than the control group, both in the set of the task and in the evolution of the task, with early-onset consumers obtaining the worst results. The control group and late-onset consumers started the task with similar values, and with the passage of time the control group increased their scores much more than the late-onset consumers, whereas the early-onset

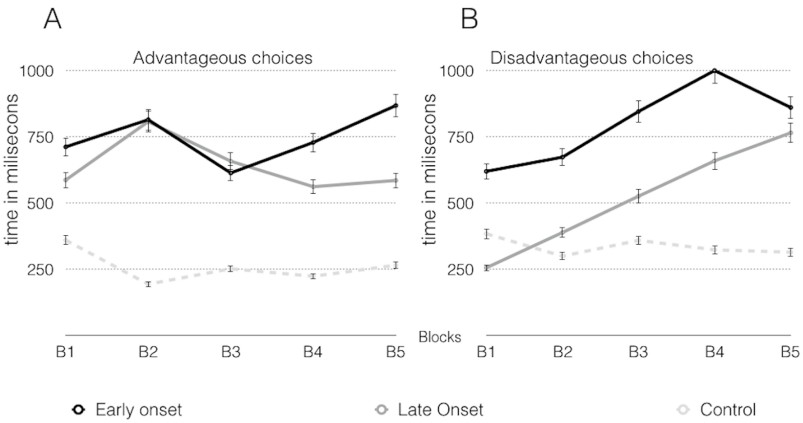

**Figure 6** Response latency in advantageous choices (A) and disadvantageous choices (B) across the inverse task.

consumers showed lower percentages of advantageous choices. The performance of the early-onset consumers was worse, which indicates that the control group and the late-onset consumers were more efficient in the examination of the characteristics of the previous four decks (*Dunn, Dalgleish & Lawrence, 2006*). Performance of consumers was better in the reverse task compared to the normal task, but this finding was no found in the control group. In the PVL parameters of the classic task, no differences were observed between the early-onset and late-onset consumers. Significant differences were found in the parameters $A$, $\alpha$ and $\lambda$ between the control group and the early-onset consumers. Significant differences in $\lambda$ values were also found between the control group and the late-onset consumers. In the inverse task, there were no significant differences in any of the PVL parameters.

The performance of the control group showed a favorable pattern of the decision-making process in the classic task, whereas this increase was lower in the consumers. The early-onset consumers always made more disadvantageous choices than advantageous. However, the late-onset consumers showed more advantageous choices from the third block, although choices were always lower than those of the control group. These results are congruent with other works that found a relationship between drug use and alterations in the decision-making process (*Ahn et al., 2008*, *2011*, *2014*; *Alameda-Bailén, Paíno-Quesada & Mogedas-Valladares, 2012*; *Bechara & Damasio, 2002*; *Bechara, Dolan & Hindes, 2002*; *Bechara et al., 2001*; *Bolla et al., 2005*; *Grant, Conttoreggi & London, 2000*; *Mogedas-Valladares & Alameda-Bailén, 2011*; *Vélez, Borja & Ostrosky-Solís, 2010*; *Whitlow et al., 2004*).

It can be concluded that the control group "learned" how the test works, and from the first block they showed their preference for the advantageous decks. On the contrary, cannabis users spent more time trying to distinguish the positive-negative characteristics of the different decks, probably due to their difficulty in using emotional signals when they must evaluate different response options. This limitation makes the task of assessing the positive-negative effects of the choices more difficult, causing either hypersensitivity to instant reward or insensitivity to punishment, according to *Bechara & Damasio (2002)*

and *Damasio (1994)*. Our results coincide in revealing hypersensitivity to instant reward (insensitivity to punishment) in late-onset consumers, although we also observed insensitivity to future consequences in early-onset consumers, which may be related to work memory or impulsiveness alterations. On the other hand, we did not find reward hypersensitivity in any of the groups (*Alloy et al., 2009*; *Nusslock & Alloy, 2017*).

Other authors suggest that the above mentioned decision-making alterations are due the difficulty to establish stimulus-reward relationships or to eradicate previously learnt responses. That is, consumers have problems reverting former learning, which could modify or eradicate responses to environmental contingencies that were previously rewarded (*Maia & McClelland, 2004*; *Rolls, 2004*). This limitation may explain the different evolution of the choices of the disadvantageous decks that offer gains at the beginning but imply losses in the long term. Thus, the control group could have modified their initial perception as of the first block, and the late-onset consumers as of the third lock, but this did not occur in the early-onset consumers. This shows that consumers, especially those of the early-onset group, either did not adequately identify the characteristics of the decks (*Fernie & Tunney, 2006*; *Lin et al., 2007*; *Lin, Chiu & Huang, 2009*) or they had problems eradicating their initial preference for the disadvantageous decks (A-B).

Our findings are related to the characteristics of the decision-making process in the PVL model (Table 4). In this sense, early-onset consumers showed less consistency, that is, lower *c* values, which implies higher randomness of choices and more influence of the previous choice, as revealed by the highest values of *A* (*Ahn et al., 2008*; *Erev & Barron, 2005*). Moreover, regarding $\alpha$ values, the early-onset consumers showed a large magnitude of gains-losses, whereas frequency was more important in the control group, and the late-onset consumers were in an intermediate position. *Alameda-Bailén et al. (2014)* and *Fridberg et al. (2010)* have reported similar results. With respect to the $\lambda$ values, we found some significant differences between groups, with lower values in the consumers. That is, the early-onset consumers perceived the losses as a neutral element, the late-onset consumers considered the gains and losses as equals, and in the control group there was loss aversion, which is partly coherent with previous findings (*Fridberg et al., 2010*) because *Alameda-Bailén et al. (2014)* obtained the opposite results, that is, loss aversion in consumers but not in the control group.

To sum up, through the PVL model, we can distinguish the different cognitive mechanisms of this model in the decision-making process. Thus, the decision-making process of the early-onset cannabis consumers was more influenced by the magnitude of the gains/losses, more determined by the short-term results, and without loss aversion. On the contrary, the performance of the control group was determined by the frequency of gains/losses, the long-term influence of the results, and loss aversion. These results show quantitative differences regarding the cognitive mechanisms, essentially between the different types of consumers, because they reveal worse results in the early-onset consumers than in the late-onset consumers. In addition, our results indicate quantitative and qualitative differences between the two types of consumers and the control group.

We also have observed that performance of the consumers was better in the reverse procedure than in the normal procedure, whereas the control group performed the reverse task worse than the classic task. However, we found few differences between groups. These findings coincide with those of other studies (*Alameda-Bailén, Paíno-Quesada & Mogedas-Valladares, 2012*; *Alameda-Bailén et al., 2014*; *Mogedas-Valladares & Alameda-Bailén*).

In the PVL parameters of the reverse task, no significant differences were found between groups. Therefore, the decision-making processes in both the consumers and the control group seem to be determined by the outcomes feedback and a random sensitivity choice, and in these processes more weight is granted to past outcomes and more sensitivity is shown to losses compared to gains.

Unlike the proposals of *Bechara, Tranel & Damasio (2000)*, and *Gordillo et al. (2010)*, the analysis of the choices made with respect to response times showed insensitivity to future consequences in the early-onset consumers, due to their preference for the disadvantageous decks, suggesting that, in this group, there are some difficulties to use immediate prospects as a guide. Late-onset consumers showed a preference for the favorable decks, with high response times, and thus they expressed hypersensitivity to reward, that is, the perspective of receiving a large, instant reward was higher than any chance of obtaining a future loss. The control group showed a preference for favorable decks with low response times, which suggests insensitivity to punishment and that the prospect of large losses does not offset any chance of gain.

Recent studies support the idea that cannabis use during adolescence can lead to more serious and long-lasting consequences. The cannabis use may affect the endocannabinoid system, which plays an important role in the brain development. This system could be altered by exogenous cannabinoids that affect its processes, specifically the development of white matter (*Solowij et al., 2011*) and synaptic pruning (*Bossong & Niesink, 2010*). Both processes are considered critical for the brain development during adolescence. The endocannabinoid system inhibits the release of glutamate, and this process is modified by the action of exogenous cannabinoids, causing alterations in synaptic pruning and interrupting the prefrontal cortex development. This effect would lead to a general disinhibition of the prefrontal cortex functions, which, indeed, shares many of the cognitive disorders related to long-term cannabis use. Depending on the dose and the duration, cannabis use could also contribute to the development of psychosis or schizophrenia (*Bossong & Niesink, 2010*; *Caballero & Tseng, 2012*). This possibility is congruent with some results indicating that the deficit in the decision-making process may be related to consumers' unusual functioning of the prefrontal cortex regions, especially the orbitofrontal cortex (*Bechara, 2003*; *Fridberg et al., 2010*; *Whitlow et al., 2004*).

## CONCLUSIONS

Cannabis consumption in adolescence should be considered a public health problem, extending to different fields, ranging from psychosis (*Di Forti et al., 2014*) to

neurocognitive disorders such as learning and memory, executive functions, attention, impulse control disorders, etc. (*Crean, Crane & Mason, 2011*; *Fontes et al., 2011*; *Gruber et al., 2012*; *Gruber, Rogowska & Yurgelun-Todd, 2009*; *Pope et al., 2003*; *Solowij & Battisti, 2008*; *Solowij et al., 2002*). In addition, the damage produced by cannabis use in the teenage brain might be permanent, that is, some altered cognitive functions might not get to recover after the withdrawal of consumption (*Meier et al., 2012*).

The evidence of the adverse and irreversible effects of cannabis in the teenage brain may have social implications (*Lubman, Cheetham & Yücel, 2015*). Early intervention programs to reduce the prevalence and delay the onset of cannabis use among teenagers are needed. Active policies are needed in order to combat the trivialization of cannabis use, basically during adolescence because the effects of cannabis on the developing brain are not comparable to the effects of this drug on a mature brain, considering that cannabis modifies the neuromaturation and, therefore, the consequences could be irreversible.

According to the results of the present study and other studies, future investigations with larger samples seem necessary to address aspects such as time and quantity of consumption, withdrawal periods, number of relapses, and poly-consumption.

Related to the task, it would be important to have more reliable and less predictable measures, adjust the scheduling of the decks (*Alameda-Bailén, Paíno-Quesada & Mogedas-Valladares, 2012*; *Alameda-Bailén et al., 2014*; *Contreras et al., 2008*; *Mogedas-Valladares & Alameda-Bailén, 2011*, *Van den Bos, Houx & Spruijt, 2006*), use presentations with more intuitive stimuli (*Gordillo et al., 2010*), and analyze the type of instruction (*Balodis, MacDonald & Olmstead, 2006*; *DeDonno & Demaree, 2008*, *Fernie & Tunney, 2006*).

As for the probabilistic models, it would be interesting to expand with the PVL simulation processes, as for example the PVL-delta (*Ahn et al., 2008*), the PVL-decay (*Ahn et al., 2011*, *2014*), and the most recent Value-Plus-Perseverance, VPP (*Worthy, Pang & Byrne, 2013*). It would also be of interest to analyze neuroanatomical aspects of decision making (*He et al., 2016*; *Polaina et al., 2015*).

Finally, highlighting the limitations of this work, as we discussed earlier, increasing the size of the sample, controlling withdrawal periods, number of relapses or poly-consumption are important aspects, but perhaps the greatest limitation of this type of work is that it does not clarify the circular relationship that is established between the decision making and the consumption of substances, since doubt remains as to whether the problems in the decision making are due to the neurotoxicity of the drugs or problems in the decision making explain the beginning of consumption, an aspect that could only be elucidated with longitudinal studies.

### Funding

The authors received no funding for this work.

## Competing Interests

The authors declare that they have no competing interests.

## Author Contributions

- Jose Ramón Alameda-Bailén conceived and designed the experiments, performed the experiments, analyzed the data, contributed reagents/materials/analysis tools, prepared figures and/or tables, authored or reviewed drafts of the paper, approved the final draft.
- Pilar Salguero-Alcañiz conceived and designed the experiments, performed the experiments, analyzed the data, contributed reagents/materials/analysis tools, prepared figures and/or tables, authored or reviewed drafts of the paper, approved the final draft.
- Ana Merchán-Clavellino conceived and designed the experiments, performed the experiments, analyzed the data, contributed reagents/materials/analysis tools, prepared figures and/or tables, authored or reviewed drafts of the paper, approved the final draft.
- Susana Paíno-Quesada conceived and designed the experiments, performed the experiments, analyzed the data, contributed reagents/materials/analysis tools, prepared figures and/or tables, authored or reviewed drafts of the paper, approved the final draft.

## Data Availability

The raw data are provided in a Supplemental File.

## Supplemental Information

Supplemental information for this article can be found online at http://dx.doi.org/10.7717/peerj.5201#supplemental-information.

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
