# Peer review of "Age of onset of cannabis use and decision making under uncertainty"

_PeerJ, doi:10.7717/peerj.5201_

## Round 0.1 · original submission · Minor Revisions

Thank you for submitting this work. I would like to bring to your attention the comments from the reviewers. In particular the methods, and the questions regarding the results. Reviewer 2 suggests that a justification for the performance data reported is added, I concur this would strengthen the paper. Make sure that you are reporting the correct analysis and as mentioned by Reviwer 1 be careful in over emphasizing the non significant comparison between the control group and the late onset users. I look forward to receiving a revised copy of the manuscript.

·

Basic reporting

The research is interesting, but the paper is hard to follow at times, particularly in the discussion.

Be consistent with tense.

There are several grammatical errors throughout the paper.

A lot of sentences have the same structure and makes it difficult to follow (e.g. a lot of sentences start with “this” and “these”).

Please put what the conditions on the x axis stand for in the figure caption.

In the intro and discussion, there is mention of previous research that looked at addiction. The research in this paper did not examine addiction. I understand the inclusion of addiction research, but it was a bit confusing at times how it relates to you research. Cannabis use, regardless of onset, does not equal addition to cannabis. Also, when discussing addiction in the intro, please discuss what the participants were addicted to in these studies.

Line 52: Do not use the word “proved” the researchers “suggested” or “found”

Line 60: what substances?

Line 86: “hat”

Line 118: why is this an exploratory study? Is it not a study? Exploratory suggests that that the task was not designed to address a question, but you are just exploring other results. Is that the case in this study?

Lines 134-137: informed consent was repeated 3 times

Line 289: “In No significant…” ?

Experimental design

I think the primary research is within the aims and scope of the journal.

I have a few comments on the Method section

Did you control for last use to ensure you are looking at residual use and not acute?

Line 162: “participants without addictive problems” what does this refer to?

I commend the authors for the way they controlled for length of use.

Line 195-196: Did you correct for multiple comparisons when using multiple t-tests (e.g. Bonferroni)? Why not use an ANOVA?

Line 197: a repeated measures analysis of variance (ANOVA)?

Validity of the findings

Results
Lines 200-205: Why start by clamming that control group made more advantageous choices than late onset group when the difference was not significant?

Line 226: Be consistent in spacing when reporting stats

Line 263: why not run some stats to see if the scores between normal and inverse were different?

Line 275-276: Shouldn’t be doing a Bonferroni if the ANOVA was not significant.

Line 289: “In No significant…” ?

Discussion

Line 314: “probably due to their inability to use emotional signals…” seems like a bold clam, did you test this?

Line 322: Who has problems? do not rely on pronouns as much throughout the paper

Lines 330-333: Expand on this

I would like to see a discussion of the limitations of the current study.

Reviewer 2 ·

Basic reporting

The paper is well written and taken care of with an adequate level of English language, clear and unambiguous.

However, there are aspects that can be improved. For example, in the explanation of the PVL model the formulas are redundant, and are already exposed in various papers, which can be accessed, and which are duly referenced. It would be better to explain the parameters, the logic of the model and omit superficial aspects “here”.

The references to the papers of Alloy, seem a little forced, since these works are based on very different tasks to igt and administered to a different clinical group.

In Table 4 & Table 5 the column “range” must be resized.

Finally, in the final section, lines 394-404, I think it should be rewritten, since it includes proposals related to this work, the age of initiation to cannabis use, with others more related to the task itself, which I think do not fit now. . In addition, the proposal that makes use of economic models of dual response difficult to fit with the IGT as it is currently configured.

The structure complies with the PeerJ standards.

The figures are adequate and quality. They are duly referenced in the text and properly titled

The raw data supplied are those used

Experimental design

This paper is within the scope of the magazine. The research objective is well defined, and is relevant, in addition, provides data in a field little studied. Research carried out according to a high technical and ethical standard. Described methods with sufficient detail and information to replicate.

Only note that the authors choose to report the percentage of elections and not the IG, although it is true that they are parallel data, which would not change the results, the usual in these works is to take into account the IG (ig = ( c + d) -a + b)). On the other hand, it would be interesting to consider alternative proposals, such as analyzing separately the elections associated with frequent losses (c-a) against the elections associated with occasional losses (d-b)

Validity of the findings

The data is relevant, statistically well worked and controlled. They abound on an interesting topic such as cannabis use and its repercussions on brain development and the dangerous trivialization of cannabis use and the importance of the development of intervention programs to cancel or delay their consumption as much as possible. The authors are also aware of the circular relationship that is established between substance use and decision making, which can be cause and effect at the same time. However the conclusions are well expressed, linked to the stated objective.

Additional comments

I agree with that Cannabis consumption in adolescence should be considered as a public health problem, therefore, it is important to have jobs that objectively show the consequences of their use, especially in adolescents, where their consequences may be permanent in terms of cognitive functions.

For all these reasons, I consider that this paper is highly recommended for publication in PeerJ, and I propose that the authors take into consideration the proposals made, since they involve making minor changes.

---

## Round 0.2 · Minor Revisions

I agree with Reviewer #1 that this change needs to be made before I can make my final decision. It is clear in the literature that emotion processing maybe impaired but this does not necessarily mean that emotion processing is impossible as implied by the word inability.

·

Basic reporting

Satisfactory

Experimental design

Satisfactory

Validity of the findings

Satisfactory

Additional comments

In line 314 you stated “probably due to their inability to use emotional signals…” Your rebuttal was not satisfactory. The word "inability" suggests that cannabis users CANNOT use emotional signals. Please change "inability" to "difficulty" which more accurately reflects the results.

---

## Round 0.3 · accepted · Accept

Thank you so much for working with me on the changes. I would like to suggest that when rebutting reviewers it is always appreciated when rebuttals are written with a positive acknowledgement and in a less brief style of writing.

#